# A Retrospective Analysis of Clinical and Epidemiological Aspects of Parvovirus B19 in Brazil: A Hidden and Neglected Virus Among Immunocompetent and Immunocompromised Individuals

**DOI:** 10.3390/v17030303

**Published:** 2025-02-22

**Authors:** Arthur Daniel Rocha Alves, Luciane Almeida Amado

**Affiliations:** Laboratório de Desenvolvimento Tecnológico em Virologia, Oswaldo Cruz Institute, Oswaldo Cruz Foundation, Rio de Janeiro 21040-900, Brazil; arthur.alves@ioc.fiocruz.br

**Keywords:** Parvovirus B19, prevalence, diagnosis, infection

## Abstract

Parvovirus B19 (B19V) infection can affect individuals of all ages, both immunocompetent and immunocompromised. This infection is typically acute and self-limiting, most commonly resulting in rash diseases and acute febrile illness. However, its involvement in atypical manifestations such as chronic kidney disease and acute liver failure have also been reported. Diagnosis of B19V is rarely conducted in these populations, and available studies on its prevalence are limited, outdated, and do not accurately depict the current situation. This study describes and discusses retrospective investigations into the role of B19V in cases of rash diseases, acute febrile illness, anemia, occurring in the context of chronic kidney disease and HIV coinfection, and acute liver failure when no identifiable etiological agent was found, focusing on various populations in Brazil. This overview underscores the importance of recognizing the potential for severe B19V infection in all individuals, regardless of perceived immune status, as well as of considering the possibility of B19V concurrent infection, in both high-risk groups and healthy individuals to reduce the risk of serious complications and improve patient outcomes, by considering the inclusion of B19V in the routine of diagnosis and implementing management strategies. This study was limited by the absence of national surveillance data of B19V in Brazil and by the analyses that occurred retrospectively.

## 1. Introduction

Parvovirus B19 (B19V) is a widespread human virus transmitted mainly by the respiratory route. However, instead of respiratory symptoms, its prodromal symptoms include fever, malaise, headache, and myalgia [1]. The virus can also be transmitted via blood or pooled blood products and from a pregnant mother to her fetus [2,3]. Its tropism for red blood cell precursors is responsible for the diverse clinical manifestations of B19V infection [1], which can range from mild to more severe complications, depending on the immunological and hematological status of individuals. Most of the illness from B19V infection in immunocompetent individuals happens in childhood and is mild, usually in the form of erythema infectiosum. Illness in adulthood may also present rash in addition to transient polyarthropathy that may happen for months to years without joint erosion. Half of adult patients with these joint symptoms may have transient rheumatoid factors and could be diagnosed for rheumatoid arthritis [1,4].

Although the infection is generally self-limited and prognosis is quite good, B19V can present serious health risks to certain populations, including pregnant women, individuals with compromised immune systems, and those with specific blood disorders [5]. In pregnancy, B19V infection is a serious complication; it can cause both miscarriage (death before week 22) and intrauterine fetal death thereafter [6]. Hydrops fetalis, massive edema in the fetus associated with death in the uterus or at birth, is best characterized, and B19V is important in the consideration of the differential diagnosis of nonimmune hydrops [7]. Patients in hematology clinics can suffer a specific complication of B19V, transient aplastic crisis (TAC), the first disease associated with B19V, that is recognized as a life-threatening acute event in individuals with sickle cell disease. This condition causes increased destruction of red blood cells due to chronic hemolytic anemia. In such cases, a B19V infection can lead to a rapid worsening of anemia. Red blood cell transfusions are typically effective in treating transient aplastic crises; however, anemia can become severe and even fatal if transfusions are unavailable or not administered promptly [5]. B19V infection can persist in individuals with weakened immune systems, leading to pure red cell aplasia, as their immune systems fail to generate an adequate neutralizing antibody response [1].

Furthermore, B19V can infiltrate and persist in various non-erythroid tissues. Its presence, detectable through viral DNA or proteins, has been linked to several uncommon diseases, such as acute and chronic inflammatory cardiomyopathies [8], rheumatoid arthritis [9], vasculitis [10], meningoencephalitis [11], hepatitis [12,13,14], and kidney disease [15]. However, it has yet to be established whether non-erythroid cells or tissues support productive viral DNA replication and the release of new virions [16] (Figure 1).

The clinical diagnosis of B19V infection is difficult due to the similarity between B19V infection signs and symptoms and other typical rash manifestations [17]. The gold standard for the laboratory diagnosis of B19V infection is the detection of specific IgM and IgG antibodies in serum samples. Commercial tests generally have sensitivities ranging from 90 to 100% and specificities ranging from 70 to 96% [18,19,20]. Recent studies recommend correlating molecular and serological tests for an accurate laboratory diagnosis of B19V infection [21]. This recommendation is due to the possibility of false-negative results since antibodies against the viral proteins may be complexed with viral particles and consequently become undetectable in serological assays [22]. Furthermore, in the case of persistent virus infection, it is important to determine the viral load and the presence of B19V-specific antibodies [16].

The molecular diagnosis for B19V infection is especially important for patients with TAC or in the immunological window and for immunosuppressed patients with persistent anemia as they do not present an adequate humoral immune response amenable to serological diagnosis [17] or B19V genotyping. For these patients, total or partial genome sequencing should also be performed [23]. A limitation of these molecular detection methods is the impossibility of distinguishing between the presence of infectious viral particles (virions) and that of noninfectious capsid-free genetic material (“naked DNA”). To overcome this limitation, several research groups have been working on removing naked DNA, with the aid of enzymes such as Benzonase^®^, before DNA extraction [24,25,26,27]. In previous studies, we aimed to evaluate this method in patient sera samples; we found that most of the samples, previously identified as positive, became negative after pretreatment. However, decreased viral DNA loads were observed in four patients, indicating that these samples contained the infectious virus. Therefore, the test was enabled to discriminate “naked DNA” from virion B19V DNA, and it can be used to clarify the role of B19V as an etiological agent associated with uncommon clinical manifestations [27].

Despite the potential impact of B19V infection in both immunocompetent and immunocompromised individuals, the infection is still neglected, underdiagnosed, and non-notified in Brazilian Diseases Information Systems. In addition, as most of the clinical and hematological manifestations of B19V-infected patients were like classical arbovirus (e.g., dengue) presentations such as fever, headache, myalgia, and retro-orbital pain, as well as rash diseases, such as measles and rubella, B19V infection is usually hidden by other epidemic viruses. Therefore, it is possible to note the absence of recent data on the prevalence of B19V in the Brazilian population, which is urgently demanding updated epidemiological studies. Then, robust surveillance measures and expanded differential diagnosis strategies are crucial to mitigate the potential clinical impact of B19V infection.

The hypothesis of this review is that there is a need to include B19V in the differential diagnosis of viral infections in specific population groups, as it is an important etiological agent of severe anemia. In this review, we present a retrospective analysis of the involvement of B19V in diverse diseases such as acute liver failure, chronic kidney disease, HIV, rash, and febrile disease among Brazilian individuals. Our findings could enhance hospital and laboratory vigilance by increasing awareness of this circulating pathogen, which may be overlooked.

## 2. Epidemiology of Parvovirus B19 in Brazil

### 2.1. Seasonal Pattern of the Infection

The seasonal pattern of the infection may differ between countries. Outbreaks occur mostly in the winter and spring, every four to five years. Nonetheless, the recent COVID-19 pandemic has disrupted the typical pattern of the B19V epidemic cycle, which was absent for three consecutive years of the pandemic. The post-2019 period showed a significant decrease in both cases and positivity rate, likely influenced by the COVID-19 pandemic and related containment measures. However, a sharp increase was observed in late 2023 and early 2024, suggesting a possible return to pre-pandemic patterns [28]. The re-emergence of B19V after the COVID-19 pandemic, at a level equal to or higher than the pre-pandemic period, has been described in several countries such as France [29], the Netherlands [30], Serbia [31], and the United States [32]. This is due to a decline in population immunity due to the social distancing required during the pandemic, creating a pool of susceptible individuals [33].

In Brazil, B19V has also exhibited the same cyclical pattern of outbreaks every two to five years, mainly during late winter and spring (from August to December), with sporadic cases during the first half of the year. However, in the Brazilian Amazon (North region), where the climate is tropical rainforest, an increase in B19V cases was reported during the autumn (between March and July) [34].

### 2.2. Prevalence of B19V Infection

B19V has a worldwide distribution; it exclusively infects humans regardless of age. However, the prevalence of anti-B19V antibodies in the population varies according to age, increasing from 2–20% in children under 5 years to 15–40% in children and adolescents aged 5–18 years, and 40–80% in the adult population [1]. In healthy adult individuals, the frequency of anti-B19V IgG antibodies in different countries varies as follows: 44% in Chile [35]; 44.1% in the Czech Republic [36]; 50% in India [37], the United States [38], and Japan [39]; 51.2% in Spain [40]; and 60–70% in England and Wales [41,42].

In Brazil, most of the data have been obtained from studies of B19V outbreaks, case reports, and investigations among small numbers of individuals [43,44,45,46,47,48,49,50,51,52,53,54,55,56,57,58,59,60,61,62,63,64,65,66,67,68,69,70,71,72,73,74,75,76,77,78,79,80,81,82,83,84,85,86,87,88,89,90,91,92,93,94,95,96,97,98,99,100]. Among blood donors, a prevalence of 50–60% has been reported for 2008–2019 [77,78,81,90]. Although antibodies are prevalent in the general population, viremia or the presence of viral DNA is rare, varying from 0% in 2012 [77] to 4% in 2008–2013 [70]. In our recent study, B19V DNA was detected in 3.32 (95% CI 1.00–7.81) per 1000 blood donors from Rio de Janeiro between 2018 and 2019. The statistical predictive model estimated that 880 (95% CI 355–2759) blood bags could be infectious during this period [100]. The high incidence detected among blood donors suggests that 2018–2019 was an epidemic period for B19V in Brazil [91,92,93,94,95,96,97,98,99,100], as has also been reported in different countries [28,101,102,103,104]. Data from the post-COVID-19 pandemic period, from January to December 2024, have shown a marked decrease in B19V DNA-positive blood donations from Rio de Janeiro to 0.08% (1/1200) donations [unpublished data].

The studies carried out in Brazil have shown regional epidemiological differences, with a high prevalence in the urban area of Belém (North region) (43%) and in Rio de Janeiro (Southeast region) (72%) [89,90,91,92], and much lower seroprevalence in some isolated areas, such as Amazonian Amerindian communities, with adult seroprevalences of only 4 to 10% [13]. A study in Niterói-RJ found a 32% prevalence among patients with exanthematous disease [93]. Therefore, these different prevalence rates may be associated with the period of occurrence of each study, with higher prevalence rates in possibly epidemic periods in Brazil: 1988–1989, 1994–1995, 1999–2000, 2004–2005, 2009–2010, 2013–2014 (Figure 2a) [34,43,44,45,46,47,48,49,50,51,52,53,54,55,56,57,58,59,60,61,62,63,64,65,66,67,68,69,70,71,72,73,74,75,76,77,78,79,80,81,82,83,84,85,86,87,88,89,90,91,92,93,94,95,96,97,98,99,100].

For immunocompromised patients, studies are scarce and are often limited to case reports. High seroprevalence of B19V has been reported in HIV patients from Rio de Janeiro between 1997 and 1998 (91%) [62] and in hemophiliac patients in São Paulo between 2011 and 2012 (78%) [81]. In these patients, B19V infection is ultimately more severe, with more episodes of severe anemia that may even require blood transfusions. These data highlight the importance of including B19V in the differential diagnosis flowchart, especially in patients with HIV, anemia and sickle cell anemia, thalassemia, leukemia, chronic kidney disease, and acute liver failure (Figure 2b). Regarding pregnant women, the prevalence in the Southeast region ranged from 54% to 71% [in São Paulo (2000–2001), and Rio de Janeiro (2003–2004) states, respectively] [59,69]; in the Midwest region, it ranged from 9% to 61% [in Goiás (1988–1989), and Federal District (2015) states, respectively] [46,87]; in the North region, the prevalence was 84% (Pará state between 1995 and 1996) [53]. Although the state of Goiás showed the lowest prevalence, it had the highest detection of acute cases (incidence) of 27% among all the states. The prevalence among this population was high, but this may reflect the period in which the studies were carried out. There is no recommendation for universal screening of pregnant women during prenatal care for B19V infection, but fetal identification for the correct management has been shown to be a strategy for improving perinatal outcomes [105].

The available data about B19V in Brazil show higher prevalence in Rio de Janeiro and São Paulo states (Southeast region), both in immunocompetent and immunocompromised individuals. This is probably due to the greater number of epidemiological investigations and laboratory diagnostics in these two largest metropolises in Brazil.

### 2.3. Molecular Epidemiology of B19V Infection

Molecular epidemiology studies of B19V have shown that the distribution of the genotypes (1 to 3) depends on the geographic origin, population age group, and type of sample analyzed. Regarding geographic origin, genotype 1 is prevalent throughout the world, whereas genotype 3 seemed to be restricted to Africa but has already been detected in countries outside this region, including Brazil. Genotype 2 viruses were detected in patients born before 1973 and not found again until recently, when it was detected in a blood donor in Germany [70,106]. In tissue samples, the prevalence of B19V genotype 1 ranges between 28 and 81% [107,108]; that of genotype 2, between 8 and 71% [65,102]; and that of genotype 3, between 0 and 50% [106,107]. There is a worldwide predominance of genotype 1, and the spread of genotype 3b worldwide has been suggested since 2009 [109].

In Brazil, the three genotypes have already been detected, and genotype 1 is the predominant one, as in other countries [54,55,64,70]. Genotype 2 was only detected in São Paulo in 2004, in bone marrow samples from a patient born in 1936 with cytopenia of unknown origin and hematological neoplasms [70]. Genotype 3b was detected in several studies; first, in the serum of patients with rash, arthropathies, and hematological disorders in Belém, Pará [54]; then, in a patient with acute anemia after a kidney transplant [55]; and, during an epidemic of EI in Niterói, Rio de Janeiro, in a patient infected with HIV [66]. Coinfection between genotypes 1a and 3b has been described in bone marrow samples from patients with leukemia [74].

## 3. Parvovirus B19 Association with Typical Clinical Manifestations

Parvovirus B19 (B19V) is associated with multiple organ involvement and various diseases, making it crucial to investigate B19V as a differential diagnosis for a wide range of clinical manifestations when no etiological agent has been identified. This investigation is essential to ensure adequate patient treatment and to enhance surveillance systems for this infection. This study describes and discusses retrospective investigations of the role of B19V in cases of rash diseases, acute febrile illness, anemia in the context of chronic kidney disease and HIV coinfection, and acute liver failure where no etiological agent was identified, focusing on different populations in Brazil.

### Patients with Rash Diseases and Acute Febrile Illness

Several diseases are known to cause rash diseases (RD) and acute febrile illness (AFI), among which the measles virus, rubella virus, dengue virus, chickenpox virus, cytomegalovirus, Epstein–Barr virus, human herpesvirus 6, human herpesvirus 7, enterovirus, Parvovirus B19, Chikungunya virus, and Zika virus stand out [22,110]. In an attempt to organize these rashes, erythema infectiosum (EI) was classified as the fifth disease.

EI is the most common clinical manifestation of B19V infection in children; it is characterized by medium-intensity facial erythema involving the cheeks—with an erythematous, edematous, and confluent appearance, commonly referred to as “slapped cheek”—starting about 18 days after infection. The rash of EI, which originates from the deposition of immune complexes in blood vessels, is generally an indicator that the patient is adequately clearing the infection and is no longer contagious [111,112].

Prodromal symptoms often go unnoticed and may include fever, headache, and nausea. A second infection phase consists of a lacy erythematous rash involving the trunk and limbs, which can be quite pruritic, and occurs 1 to 4 days after the initial erythema. The rash usually disappears within 1 week, but it can be transient or recurrent and occur weeks to months later after being induced by certain environmental factors, such as exposure to sunlight, hot temperatures, and exercise [113,114]. Other symptoms associated with EI may include itching, vesicles, and scaly dermatitis [4].

In a tropical country like Brazil, where there is a triple epidemic of arboviruses (i.e.*,* dengue, Zika, and Chikungunya) with the frequent emergence of new arboviruses such as Mayaro and Oropouche, EI can be hidden by other diseases and B19V outbreaks are not adequately diagnosed [85]. In this study, we examined B19V infection as a potential differential diagnosis among patients with RD (n *n* = 54) and AFI (*n* = 60) who had no defined etiology. These patients were admitted to hospitals or reference centers for infectious diseases in the state of Rio de Janeiro (Southeast region) between 2018 and 2019 [95]. The B19V DNA was detected in three (5.5%) RD patients, with a viral load of 10^4^ IU/mL, and in four (6.6%) AFI patients, with a viral load of 10^5^ IU/mL. The overall detection of B19V DNA in our study was lower (6.1%) than that reported in previous studies: B19V DNA positivity in 63% in São Paulo state (Southeast region) from 2013 to 2014 [85], 31.5% in Rio de Janeiro state (Southeast region) during 1994–1999 [52], and 62% in Amazonas state (North region) from 1998 to 1999 [61]. During the follow-up investigation, all consecutive samples (from RD collected at 1 month, 3 months, and 6 months and from AFI collected at 1 week, 1 month, and 2 months) were negative for B19V DNA, which indicates an acute, benign, and self-limiting infection in these patients. A clinical analysis of these patients has shown that all B19V-positive patients had anemia (OR = 73.0, *p* = 0.007, CI = 3.09–1720.1 for RD and OR = 10.6, CI = 1.1–112, *p* = 0.04 for AFI), and it could be severe if it happens in immunocompromised individuals and those with underlying hematological issues. This finding underscores the need for effective surveillance and considering the differential diagnosis of B19V in cases of RD and AFI, especially when anemia is present. (Figure 3, Table 1).

## 4. Parvovirus B19 Association with Uncommon Clinical Manifestations

### 4.1. Patients with Chronic Kidney Disease

Glomerulonephritis is the edema in the glomeruli that can lead to chronic kidney disease (CKD), a condition that arises from several diseases (e.g., diabetes mellitus and hypertension), which leads to a gradual loss in the rate of glomerular filtration and alters the functioning of the kidney [15,117]. CKD patients are usually submitted to hemodialysis, and kidney transplantation. Infectious diseases are the second leading cause of death in CKD patients, and patients who undergo hemodialysis have a greater risk of blood-borne viral infections, such as B19V [117].

Infection with B19V in these patients can result in severe anemia. CKD patients on dialysis increased their susceptibility to anemia after B19V infection due to the occurrence of deficient erythropoietin production, weakened immune response, and decreased erythrocyte survival, which can lead to serious anemia [15].

We previously reported the prevalence of B19V in three Brazilian hemodialysis units: two of them in Rio de Janeiro state in 2013 (*n* = 32) and 2014 (*n* = 61), and the other in Ceará state in 2015 (*n* = 128) [88]. The overall B19V DNA detection of the study was 65.6%, but as the period studied was different between the units, we observed a difference in B19V prevalences, which ranged from 3.1% in unit 1 to 60.6% in unit 2 and 83.6% in unit 3. It was possible to analyze the sequence data in only two samples due to the amount of viral load; both samples belonged to genotype 1a (Figure 3, Table 1). Surprisingly, other prevalences found in the literature were low when we compared to our study, that is, 10% in Iran [118] and 8.1% in Minas Gerais state, Brazil [83], but probably due to the non-epidemic period in these places.

All CKD patients showed a low viral load, with a mean of 10^4^ IU/mL. This was accompanied by the presence of IgG antibodies and the absence of IgM antibodies, suggesting a persistent infection in these individuals. However, the clinical significance of this prolonged viral clearance and low levels of viremia remains unclear and continues to be a topic of debate [88].

Although anemia in this study population was not statistically significant (*p* > 0.05), it was found in 60% of CKD patients, which is common due to erythropoietin and iron deficiency and the dialysis procedure itself [15,119]. Anemia is associated with more hospitalization, increased mortality, and worse quality of life, but morbidity and mortality depend mainly on the etiology of anemia and the stage of kidney disease [15,117]. Future studies are needed to clarify the importance of B19V infection as a cause of anemia in this population. A statistically significant correlation (*p* < 0.05) was found between anemia and B19V coinfection with hepatitis B/hepatitis C. It has been suggested that B19V infection may worsen in the presence of hepatotropic viruses, which could help explain this condition [120].

B19V DNA detection was significantly higher (OR = 28.1, CI = 13.5–58.5, *p* = 0.001) in the CKD patients under dialysis (65%) than in the control group (blood donors; 6.3%). Because of the immune deficiency condition in these patients and the routine exposure to dialysis procedures, there is a potential risk for parenterally transmitted infections. Because of this, it is important to investigate B19V in this high-risk population since persistent B19V infection may become clinically significant after renal transplantation with the administration of immunosuppressive therapy. Furthermore, this persistent viremia should be considered a potential risk for the dissemination of infection in closed hemodialysis units.

### 4.2. Patients Coinfected with HIV

In immunocompromised patients, with a reduced ability to produce a neutralizing antibody response and/or due to a persistent failure of the bone marrow to produce erythroid precursors, B19V infection can cause persistent anemia. Predisposing conditions for this immunocompromise include Nezelof syndrome [121], leukemia [122,123,124,125], myelodysplastic syndrome [126], Burkitt’s lymphoma, lymphoblastic lymphoma [127], astrocytoma, human immunodeficiency virus (HIV) infection—mainly if patients are not receiving antiretroviral therapy [98,128,129,130], bone marrow transplantation [124,125,131], organ transplantation [132,133], or cancer chemotherapy treatment [127]. Patients present low levels (or absence) of specific antibodies, and persistent viremia is detected [121]. Clinical features include fatigue and pallor, whereas immune-mediated symptoms (EI and arthralgia) are generally not present [121].

Recently, we reported three rare cases of B19V infection in adults living with HIV in Rio de Janeiro state between 2021 and 2023 [98]. All three patients presented severe anemia and required red blood cell transfusions; in each case, real-time PCR was crucial for diagnosing B19V (Figure 3, Table 1). A study conducted in Rio de Janeiro state between 2001 and 2003 estimated a frequency of B19V seroconversion of 31.8% in a cohort of 88 HIV-infected patients. It showed that patients who seroconverted were 5.40 times more likely to have anemia than those who did not [66]. There is a lack of more recent data on the frequency of B19V and HIV coinfection in Brazil.

Our findings showed that adherence to antiretroviral therapy (ART) was crucial to B19V clearance in patients with HIV and highlighted the importance of the early recognition of B19V infection in unexplained cytopenias [98]. This study emphasizes the importance of diagnosing B19V in HIV-coinfected patients. Cytopenias often have multiple causes, necessitating comprehensive clinical evaluation to prevent delayed diagnosis and morbidity. Distinguishing persistent B19V infection from other opportunistic infections in HIV-coinfected individuals poses challenges. Therefore, clinicians should maintain a high suspicion of B19V in HIV patients with cytopenias and advanced immunosuppression. Treatment decisions should be clinically guided. While intravenous immunoglobulin (IVIG) therapy clears B19V viremia, ART adherence is crucial for normalizing blood cell counts and preventing B19V relapses.

### 4.3. Patients with Acute Liver Failure

The spectrum of liver diseases associated with B19V is broad, ranging from increased transaminases to acute liver failure (ALF), and may occur in 4.1% of the patients infected with B19V [134]. Acute hepatitis and ALF have been described more frequently in pediatric groups, but the incidence is very rare [135]. Coinfection of B19V with other hepatotropic viruses may lead to severe ALF [120], which could result in severe jaundice, high levels of bilirubin and liver enzymes, and unfavorable clinical outcomes that may culminate in the death of most patients.

In a previous study, we investigated the presence of B19V in the serum and liver tissues of 30 ALF patients who underwent liver transplantation from Rio de Janeiro state during the period of 2004 to 2012 and evaluated the viral activity in infected livers by real-time PCR, serological tests, and examination of B19V mRNA (NS1 and VP1 transcripts) expression in the liver. The serum and liver samples from seven patients were positive for B19V DNA (10^3^ to 10^5^ copies/mL), and most of them presented detectable anti-B19V IgG, suggesting persistent infection. All of them presented jaundice, anemia, and anisocytosis [71,72]. This is consistent with previous studies investigating liver explants in adults with ALF in other countries [136,137,138,139]. To our knowledge, this was the first study of B19V in a Brazilian population with ALF.

During the ALF study, B19V transcripts (mRNA) were detected in all patients with liver dysfunction. These data indicate active gene expressions of structural and nonstructural proteins [140,141], which is indicative of intrahepatic B19V replication. The liver injury is a result of DNA damage caused by the NS1 viral protein, which leads to mitochondrial instability and caspase 9 activation, resulting in hepatocellular apoptosis. This provides evidence of B19V infection during the course of cryptogenic ALF and suggests a role for B19V in liver injury, leading to the worst outcomes in coinfected patients and in patients with cryptogenic ALF (Table 1, Figure 3) [71].

In agreement with previous studies, the severity of liver disease is enhanced due to the coinfection of B19V and other hepatotropic viruses (e.g., hepatitis B virus—HBV) [5,120,135]. The results indicate that B19V can cause liver injury on its own and also has a synergistic effect when coinfected with HBV, leading to a poorer prognosis. Furthermore, nearly all patients with acute liver failure (ALF) exhibited anemia and thrombocytopenia, which are typical signs of bone marrow involvement associated with B19V infection [5].

Phylogenetic analyses revealed that genotype 1a was found in three patients, while genotype 3b was identified in four. Unexpectedly, genotype 3 emerged as the most prevalent among this population. This was previously reported regarding Brazilian studies in anemic HIV-infected patients [66,68], in a patient with systemic lupus erythematosus, and children with EI from the state of Pará [65].

The mechanism by which B19V infection relates to hepatitis is not clear, but both direct and indirect—immunologically mediated—effects leading to liver damage have been suggested [142,143]. As hepatocytes can express globosides and glycosphingolipids, which are the receptors for B19V adsorption, this virus can enter hepatocytes and establish an infection [144]. Direct effects on liver damage are then exerted by inducing hepatocyte apoptosis due to the activation of caspases 3 and 9 [143,145]. B19V indirect effects on liver damage are due to an increase in the circulation of cytotoxic CD8+ T lymphocytes, which leads to a defective and altered differentiation of monocytes and macrophages, causing decreased IL-1 and increased IFN-γ and TNF-α expression, intensifying liver damage, and leading to acute hepatitis [142].

B19V contributed to liver damage among patients with acute liver failure; clinically, the worse outcomes were for patients coinfected with HBV. However, longitudinal studies are needed to demonstrate the causal relationship between B19V infection and ALF. The data demonstrate the importance of the differential diagnosis of B19V in cases of ALF, especially during epidemic periods of EI and in patients with refractory anemia.

## 5. Conclusions

These results highlight the potential role of B19V as a causative agent associated with various clinical manifestations and the need for future longitudinal studies to establish causal relationships between B19V infection and clinical outcomes. The prevalence identified in this study suggests that B19V is present in different population groups and emphasizes the importance of enhanced surveillance of this infection, as it affects both immunocompetent and immunocompromised individuals. Therefore, including B19V in the differential diagnosis is crucial for epidemiological purposes and effective patient management.

Given these considerations, Brazil needs to enhance its surveillance efforts to monitor the epidemiology of B19V. This will help in promptly identifying potential outbreaks and tracking case reports, particularly among individuals at higher risk of complications. We recommend employing strategies similar to those that have been implemented in several European countries. For instance, all blood donations in Germany and four regions in Austria are screened using a real-time nucleic acid test (NAT) for B19V [146,147]. Additionally, since 2008, Japanese Red Blood Centers have been using a chemiluminescent enzyme immunoassay to screen all donated blood for B19V antigens [148]. Furthermore, it is essential to register cases of B19V infection in Brazil’s database of the Notifiable Diseases Information System—SINAN. This system relies on medical reports of diseases that require mandatory notification. The SINAN database is a valuable source of information for national epidemiological surveillance assessment. However, B19V infection is not currently included in the list of compulsory notifications. Fully utilizing this system could enable the collection of crucial data for calculating key indicators needed to monitor this infection, and support the formulation and evaluation of health policies, plans, and programs, supporting the decision-making process and ultimately contributing to improving the health of the population. Also important is the occurrence of longitudinal studies to establish causal relationships between B19V infection and clinical outcomes, in all previously studied populations reported.

B19V is a continuous matter of interest to the biomedical community since it is associated with a wide spectrum of clinical manifestations, some of which are still controversial. In addition, due to the lack of efficiency of in vitro tissue cultures and animal models, classical virological and virus/host interaction studies involving B19V are still limited. Therefore, extended awareness of the actual pathogenetic role of B19V in human diseases and the development of prophylactic and therapeutic treatment options will continue to be relevant issues that will demand continuous efforts from virologists.

## Figures and Tables

**Figure 1 viruses-17-00303-f001:**
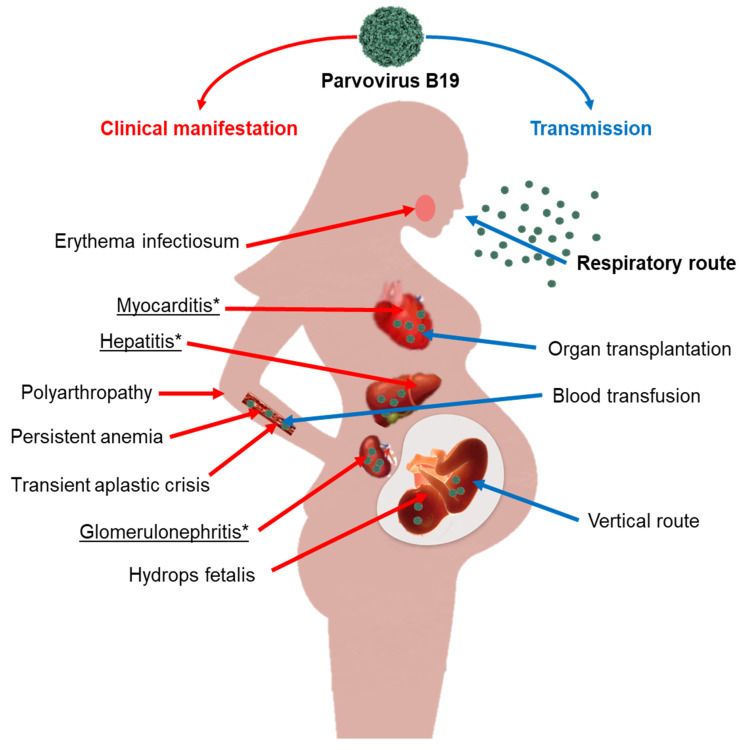
Transmission and clinical manifestations of Parvovirus B19. Those diseases marked with an asterisk (*) and underlined represent atypical diseases derived from B19V infections. The respiratory route has been marked using bold characters because it is the main transmission route of B19V.

**Figure 2 viruses-17-00303-f002:**
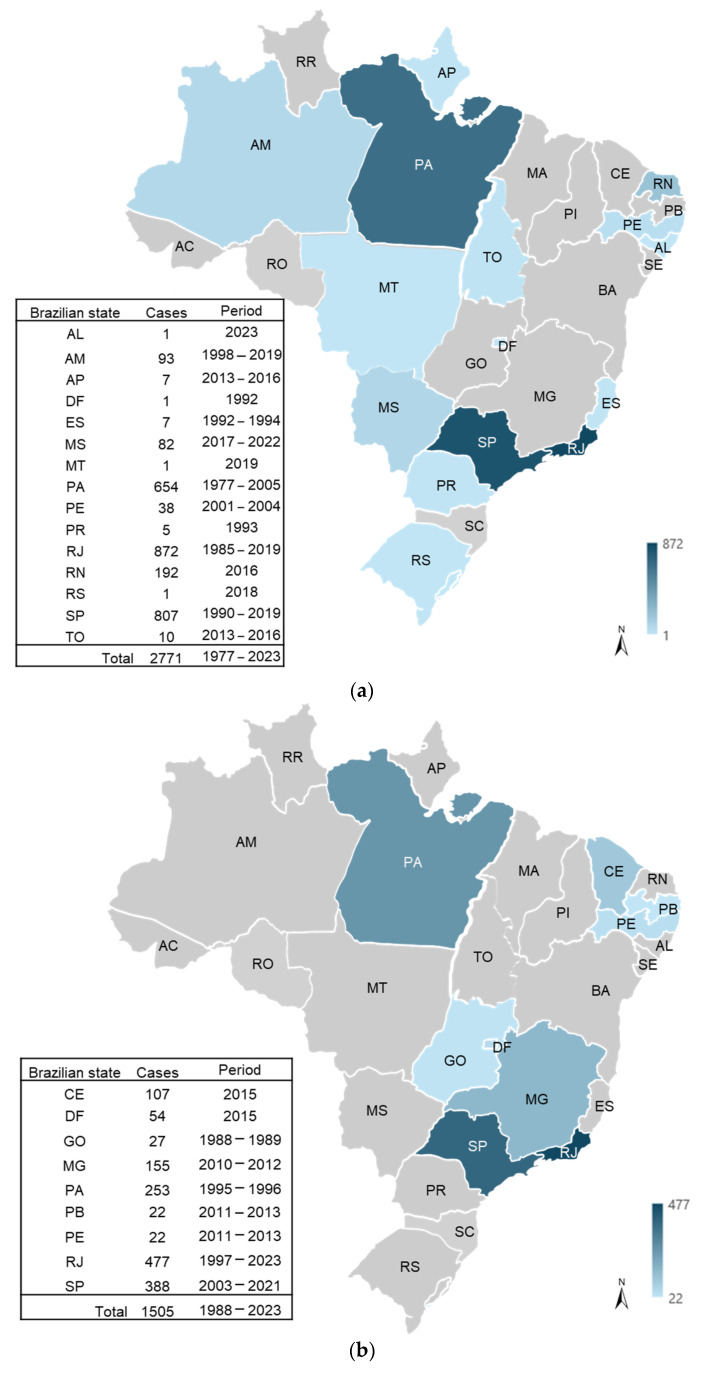
Cases of Parvovirus B19 infection in Brazilian states, according to studies published between 1978 and 2023: (**a**) cases of immunocompetent individuals with mild symptoms; (**b**) cases of immunocompromised patients and in pregnant women. The acronyms of the Brazilian states are as follows: Acre (AC); Alagoas (AL); Amapá (AP); Amazonas (AM); Bahia (BA); Ceará (CE); Distrito Federal (DF); Espírito Santo (ES); Goiás (GO); Maranhão (MA); Mato Grosso (MT); Mato Grosso do Sul (MS); Minas Gerais (MG); Pará (PA); Paraíba (PB); Paraná (PR); Pernambuco (PE); Piauí (PI); Rio de Janeiro (RJ); Rio Grande do Norte (RN); Rio Grande do Sul (RS); Rondônia (RO); Roraima (RR); Santa Catarina (SC); São Paulo (SP); Sergipe (SE); Tocantins (TO).

**Figure 3 viruses-17-00303-f003:**
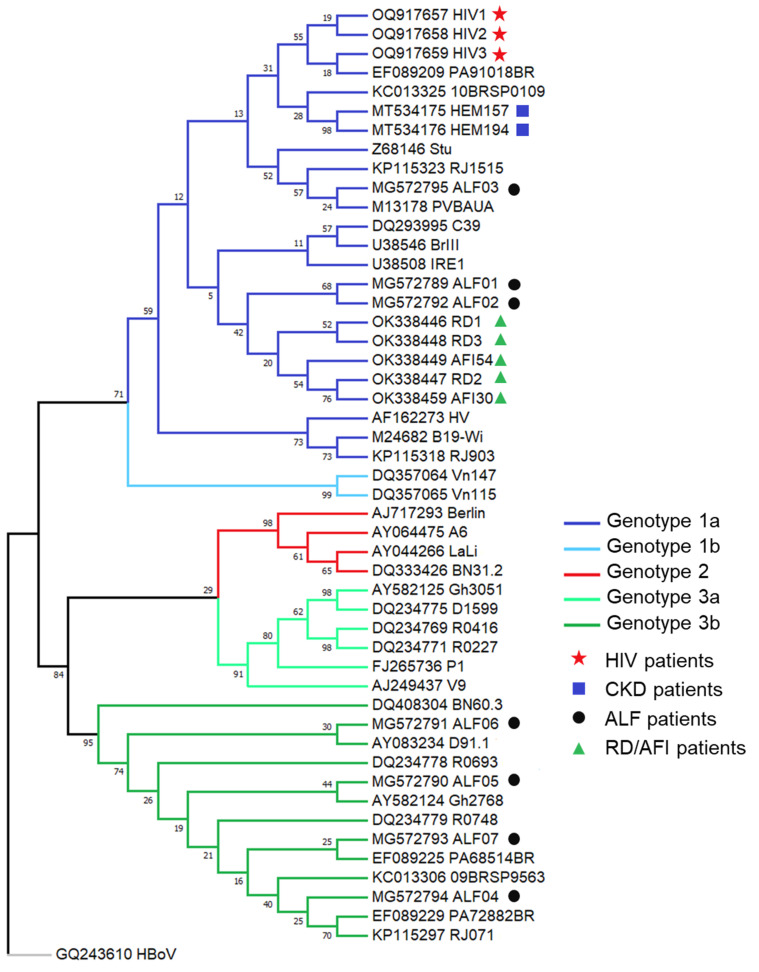
Parvovirus B19 phylogenetic tree inferred by the maximum likelihood method and Tamura–Nei model [115]. Evolutionary analyses were conducted in the MEGA program (version 7.0.26). The bootstrap consensus tree was inferred from 10,000 replicates [116]. VP1/VP2 gene (among 420 bp) in patients with acute liver failure (ALF; black circles), chronic kidney disease (CKD; dark-blue squares), rash diseases and acute febrile illness (RD/AFI; dark-green triangles), and anemia coinfected with HIV (red stars). A human bocavirus (HBoV; GQ243610) sequence was used to root the tree. All samples presented in this review were deposited in the GenBank database under the accession number presented in Table 1.

**Table 1 viruses-17-00303-t001:** Overview of Parvovirus B19 positive patients ^a^ among the studied populations.

Study Population	ID	Gender	Age	Collection Year	IgM	IgG	B19V Load (IU/mL)	Genotype	GenBank ID
Acute liver failure	ALF01	Female	9	2006	Negative	Positive	5.5 × 10^5^	1a	MG572789
	ALF02	Female	11	2005	Negative	Positive	3.2 × 10^5^	1a	MG572792
	ALF03	Female	31	2009	Negative	Positive	1.3 × 10^5^	1a	MG572795
	ALF04	Female	48	2008	Negative	Positive	2.1 × 10^5^	3b	MG572794
	ALF05	Female	49	2008	Negative	Positive	2.2 × 10^5^	3b	MG572790
	ALF06	Female	49	2004	Negative	Positive	1.3 × 10^5^	3b	MG572791
	ALF07	Female	52	2005	Negative	Positive	8.4 × 10^4^	3b	MG572793
Chronic kidney disease	HEM157	Female	70	2015	Positive	Positive	5.3 × 10^4^	1a	MT534175
	HEM194	Female	47	2015	Positive	Positive	3.8 × 10^4^	1a	MT534176
Rash diseases and acute febrile illness	RD1	Male	7	2018	Positive	Positive	4.0 × 10^4^	1a	OK338446
	RD2	Female	6	2018	Positive	Positive	2.9 × 10^4^	1a	OK338447
	RD3	Male	1	2018	Positive	Positive	7.1 × 10^4^	1a	OK338448
	AFI54	Female	22	2018	Positive	Positive	3.0 × 10^5^	1a	OK338449
	AFI30	Male	50	2019	Positive	Positive	1.3 × 10^4^	1a	OK338450
People living with HIV	HIV1	Male	27	2021	Negative	Negative	4.3 × 10^10^	1a	OQ917657
	HIV2	Male	61	2021	Negative	Positive	8.9 × 10^3^	1a	OQ917658
	HIV3	Male	27	2022	Negative	Positive	5.9 × 10^4^	1a	OQ917659

^a^ In this table, there are only positive patients by sequencing, which includes all patients with acute liver failure and people living with HIV studied populations

## Data Availability

No new data were created or analyzed in this study. Data sharing is not applicable to this article.

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
