# Peer review of "A Retrospective Analysis of Clinical and Epidemiological Aspects of Parvovirus B19 in Brazil: A Hidden and Neglected Virus Among Immunocompetent and Immunocompromised Individuals"

_viruses, 2025, doi:10.3390/v17030303_

Round 1
Reviewer 1 Report
Comments and Suggestions for Authors
This is a well written review.
Line 33: While the arthropathy is typically transient, it may persist for months or years without joint erosion. Half of adult patients with joint symptoms would meet ACR/EULAR classification criteria for a diagnosis of rheumatoid arthritis. Patients may have transient rheumatoid factor as well.
Line 93: Rubella may also mimic B19V infection presentation; this is mentioned later in the paper.
Lines 163-168: It would be instructive to stratisfy prevalence in pregnant women by number of pregnancies or live births, as a surrogate for the amount of close exposure to children.
Table 1: It would be instructive to translate the B19V viral load from IU/mL to virions. How many virions does 1 IU/mL represent?
Lines 304-308: It would be informative to know if the IgG antibodies present in the CKD patients were anti-VP2 antibody only, or anti-VP2 antibody as well, since the later is neutralizing while the former is not. A discussion of this would be helpful in paragraph in lines 330-340, as well.
Lines 378-383: B19V transcription per se does not necessarily indicate virion production. In restricted infection, VP1 and VP2 transcripts may not be produced in sufficient amounts or not at all. NS1 transcript is produced in restricted infections, e.g. in liver, which is the basis for DNA damage leading to mitochondrial instability and caspase 9 activation with resulting hepatocellular apoptosis. The transcript produced in liver cells, for NS1, VP1 and/or VP2 would inform the discussion and should be reviewed.This is a well written review.
Author Response
Comment: This is a well written review.
Line 33: While the arthropathy is typically transient, it may persist for months or years without joint erosion. Half of adult patients with joint symptoms would meet ACR/EULAR classification criteria for a diagnosis of rheumatoid arthritis. Patients may have transient rheumatoid factor as well.
Reply: Thank you for pointing this out. This information was added to the manuscript on lines 35-37.
Comment: Line 93: Rubella may also mimic B19V infection presentation; this is mentioned later in the paper.
Reply: We agree with your review, and rubella was added on line 95.
Comment: Lines 163-168: It would be instructive to stratisfy prevalence in pregnant women by number of pregnancies or live births, as a surrogate for the amount of close exposure to children.
Reply: We agree with your recommendations, but, unfortunately, it is not possible to stratify the prevalence according to pregnancies or live births due to the insufficient information regarding the articles in the literature.
Comment: Table 1: It would be instructive to translate the B19V viral load from IU/mL to virions. How many virions does 1 IU/mL represent?
Reply: According to Koppelman et al., 2004 (https://doi.org/10.1046/j.0041-1132.2004.00610.x) 1 IU/mL represents 3,34 copies/mL of Parvovirus B19 DNA, but we can not determine the amount of virions 1 IU/mL represents. The calculation of copies/mL to IU/mL was previously done and is the value stated on table 1.
Comment: Lines 304-308: It would be informative to know if the IgG antibodies present in the CKD patients were anti-VP2 antibody only, or anti-VP2 antibody as well, since the later is neutralizing while the former is not. A discussion of this would be helpful in paragraph in lines 330-340, as well.
Reply: Unfortunately, we cannot discuss anything in this regard, as the protocol of the commercial kit used (Parvovirus B19 IgG #EI2580-8601 G, Euroimmun, Brazil) does not inform which protein was used to prepare the antigen (whether VP1 or VP2), even though we believe it is VP2 as it is the most immunogenic and prevalent in the viral capsid. In previous studies, other commercial kits used (Serion ELISA classic Parvovirus B19 IgG #ESR122G, VirionSerion, Brazil) did not inform the protein used as an antigen.
Comment: Lines 378-383: B19V transcription per se does not necessarily indicate virion production. In restricted infection, VP1 and VP2 transcripts may not be produced in sufficient amounts or not at all. NS1 transcript is produced in restricted infections, e.g. in liver, which is the basis for DNA damage leading to mitochondrial instability and caspase 9 activation with resulting hepatocellular apoptosis. The transcript produced in liver cells, for NS1, VP1 and/or VP2 would inform the discussion and should be reviewed.This is a well written review.
Reply: Thank you for pointing this out. In this newly submitted version, this paragraph was reviewed and stated as follow (lines 371-378):
Lines 371-378: During the ALF study, B19V transcripts (mRNA) were detected in all patients with liver dysfunction. These data indicate active gene expressions of structural and nonstructural proteins [140,141], which is indicative of intrahepatic B19V replication. The liver injury is resulted of the DNA damage caused by NS1 viral protein that leads to mitochondrial instability and caspase 9 activation with resulting hepatocellular apoptosis. This provides evidence of B19V infection during the course of cryptogenic ALF and suggests a role for B19V in liver injury, leading to the worst outcomes in coinfected patients and in patients with cryptogenic ALF.

Reviewer 2 Report
Comments and Suggestions for Authors
This study shows the infection status of B19V in Brazil and the risk to high-risk groups such as pregnant women, HIV-infected individuals, patients with chronic kidney disease, patients with blood diseases, and immunocompromised individuals, as well as healthy individuals, and concludes by suggesting the need for a national surveillance system.
The study covered a wide range of studies, and in that it comprehensively discusses the risks of B19V, this paper is worthy of being published as a review. However, the national surveillance system proposed in the conclusion is not sufficiently discussed. Therefore, as the reviewer, I would like the authors to add the following:
1. In Europe, epidemic trends are being monitored through numerous case reports and blood transfusion testing. I would like the authors to state the need to continue epidemiological research with further reference to these reports.
2. Also in Europe, EpiPulse - the European surveillance portal for infectious diseases accepts voluntary disease reports from medical institutions. I would like the authors to mention websites and organizations that make it easy for medical institutions to report and have a system for analyzing collected data.
Author Response
Comment: This study shows the infection status of B19V in Brazil and the risk to high-risk groups such as pregnant women, HIV-infected individuals, patients with chronic kidney disease, patients with blood diseases, and immunocompromised individuals, as well as healthy individuals, and concludes by suggesting the need for a national surveillance system.
The study covered a wide range of studies, and in that it comprehensively discusses the risks of B19V, this paper is worthy of being published as a review. However, the national surveillance system proposed in the conclusion is not sufficiently discussed. Therefore, as the reviewer, I would like the authors to add the following:
1. In Europe, epidemic trends are being monitored through numerous case reports and blood transfusion testing. I would like the authors to state the need to continue epidemiological research with further reference to these reports.
2. Also in Europe, EpiPulse - the European surveillance portal for infectious diseases accepts voluntary disease reports from medical institutions. I would like the authors to mention websites and organizations that make it easy for medical institutions to report and have a system for analyzing collected data.
Reply: Thank you for your comment. We added the requested information in the conclusion section of the manuscript (lines 406-432).
Lines 406-432: Given these considerations, it is essential for Brazil to enhance its surveillance efforts to monitor the epidemiology of B19V. This will help in promptly identifying potential outbreaks and tracking case reports, particularly among individuals at higher risk of complications. We recommend employing strategies similar to those that have been implemented in several European countries. For instance, all blood donations in Germany and four regions in Austria are screened using a real-time nucleic acid test (NAT) for B19V [146,147]. Additionally, since 2008, Japanese Red Blood Centers have been using a chemiluminescent enzyme immunoassay to screen all donated blood for B19V antigens [148]. Furthermore, it is essential to notify the cases of B19V infection in the Brazil´s database of the Notifiable Diseases Information System – SINAN. This system relies on medical reports of diseases that require mandatory notification. The SINAN database is a valuable of information for national assessment of epidemiological Surveillance assessment. However, B19V infection is not currently included in the list of compulsory notifications. Fully utilizing this system could enable the collection of crucial data for calculating key indicators needed to monitor this infection, and support the formulation and evaluation of health policies, plans, and programs, supporting the decision-making process and ultimately contribute to improving the health of the population. Also, is important the occurrence of longitudinal studies to establish causal relationships between B19V infection and clinical outcomes, in all previously studied populations reported.
146. Schmidt M, Themann A, Drexler C, Bayer M, Lanzer G, Menichetti E, Lechner S, Wessin D, Prokoph B, Allain JP, Seifried E, Hourfar MK. Blood donor screening for parvovirus B19 in Germany and Austria. Transfusion. 2007;47(10):1775-82.
147. Grabarczyk P, Korzeniowska J, Liszewski G, Kalińska A, Sulkowska E, Krug- Janiak M, Kopacz A, Łetowska M, Brojer E. Badanie DNA parvowirusa B19 (B19V) u Polskich dawcoów krwi, 2004–2010 [Parvovirus B19 DNA testing in Polish blood donors, 2004–2010]. Przegl Epidemiol. 2012;66(1):7–12.
148. Sakata H, Matsubayashi K, Ihara H, Sato S, Kato T, Wakisaka A, Tadokoro K, Yu MY, Baylis SA, Ikeda H, Takamoto S. Impact of chemiluminescent enzyme immunoassay screening for human parvovirus B19 antigen in Japanese blood donors. Trans-fusion. 2013;53(10 Pt 2):2556–66.

Reviewer 3 Report
Comments and Suggestions for Authors
This retrospective analysis investigates the clinical and epidemiological aspects of Parvovirus B19 (B19V) infection in Brazil, focusing on both immunocompetent and immunocompromised individuals. B19V, a widespread human virus transmitted primarily through respiratory routes, is often associated with mild, self-limiting illnesses such as rash diseases and acute febrile illness. However, it can also lead to severe complications, particularly in high-risk groups like pregnant women, individuals with hematological disorders, and those with compromised immune systems. Despite its potential for serious health impacts, B19V remains underdiagnosed and underreported, often overshadowed by other epidemic viruses such as dengue, Zika, and chikungunya.
The study highlights the importance of including B19V in differential diagnoses, especially in cases of rash diseases, acute febrile illness, chronic kidney disease (CKD), HIV co-infection, and acute liver failure (ALF). It underscores the need for enhanced surveillance and diagnostic strategies to mitigate the clinical impact of B19V, particularly in Brazil, where the virus exhibits cyclical outbreaks and regional variations in prevalence. The research also explores the molecular epidemiology of B19V, identifying the predominance of genotype 1 in Brazil, with occasional detection of genotype 3b, which has been associated with more severe outcomes.
Specific comments:
Abstract:
The abstract provides a good summary of the study's objectives, methods, and key findings. However, it is somewhat lengthy and could be more concise. For instance, the phrase "by considering to include B19V in differential laboratory diagnosis" is awkwardly phrased and should be reworded for clarity. Additionally, the abstract does not mention the limitations of the study, which is important for setting expectations. Including a brief statement about the study's limitations, such as the lack of national surveillance data or the retrospective nature of the analysis, would provide a more balanced overview.
Introduction:
The introduction effectively sets the stage for the study by discussing the transmission, clinical manifestations, and risks associated with B19V infection. However, it could be improved by more clearly identifying the gaps in the literature, particularly in the context of Brazil. For example, the introduction could emphasize the lack of recent data on B19V prevalence in Brazil and the need for updated epidemiological studies. Additionally, the phrase "Despite the infection is general self-limited" contains a grammatical error ("is general" should be "is generally"). The introduction also lacks a clear hypothesis or research question, which would help guide the reader and provide a stronger foundation for the study.
Epidemiology of Parvovirus B19 in Brazil:
- The section is somewhat repetitive, particularly in discussing the cyclical nature of outbreaks.
- The phrase "Brazil has no national epidemiological surveillance system for B19V, and cases are not compulsorily reported" is repeated multiple times, which could be streamlined.
- The discussion on the impact of COVID-19 on B19V epidemiology is interesting but lacks depth. More data or references could be provided to support the claims.
- The section could benefit from a clearer structure, perhaps with subheadings to separate discussions on prevalence, seasonal patterns, and regional differences.
Parvovirus B19 Association to Typical and Uncommon Clinical Manifestations:
- The section is overly long and could be divided into subsections for better readability.
- The discussion on rash diseases and acute febrile illness is somewhat repetitive, particularly in the context of anemia.
- The phrase "The overall detection of B19V DNA in our study was lower (6.1%), than showed in previous studies" is awkwardly phrased and should be reworded.
- The discussion on chronic kidney disease lacks a clear conclusion on the clinical significance of persistent B19V infection in these patients.
- The section on HIV co-infection is informative but could benefit from more recent data, as the cited studies are from 2001-2003.
- The discussion on acute liver failure is thorough but could be more concise. The mechanism of liver damage by B19V is discussed in detail, but the clinical relevance of these findings could be emphasized more.
Conclusion:
The conclusion effectively summarizes the key findings and emphasizes the importance of B19V in differential diagnosis. However, it is somewhat repetitive, particularly in reiterating the need for enhanced surveillance. The phrase "underscore the importance of including B19V in the differential diagnosis" is awkwardly phrased and should be reworded. Additionally, the conclusion could benefit from a more forward-looking perspective. For example, it could discuss potential future research directions, such as longitudinal studies to establish causal relationships between B19V infection and clinical outcomes, or public health interventions to improve surveillance and diagnosis.
Figures:
Figure 3
- The tree is cluttered and difficult to read, particularly due to the large number of sequences and the small font size. The labels for the sequences are not clearly differentiated, making it hard to distinguish between different groups (e.g., acute liver failure, chronic kidney disease, etc.).
- The bootstrap values are not clearly indicated, which is crucial for assessing the robustness of the tree's branches. Without these values, the reliability of the phylogenetic relationships is questionable.
- The tree does not clearly differentiate between the different genotypes (1a, 1b, 2, 3a, 3b). A color-coded or labeled system could help in distinguishing these genotypes more effectively.
- The use of human bocavirus (HBoV) as an outgroup is appropriate, but the tree does not clearly show the divergence between B19V and HBoV, which could be improved by adjusting the scale or adding more outgroup sequences.
- The tree lacks clear annotations for key nodes or clades, which would help in interpreting the evolutionary relationships between the sequences.
- The manuscript does not provide sufficient details on the methodology used to construct the tree (e.g., alignment method, model selection, etc.). This information is crucial for reproducibility and validation of the results.
Typographical Errors:
- Page 1: "Despite the infection is general self-limited" should be "Despite the infection being generally self-limited."
- Page 2: "transient aplastic crisis (TAC). TAC was the first human illness associated with B19V" is repetitive. Consider rephrasing.
- Page 3: "correlating molecular and serological tests for an accurate laboratory diagnosis of B19V infection [21] due to the possibility of false-negative results since antibodies against the viral proteins may be complexed with viral particles and consequently become undetectable in serological assays [22]." This sentence is overly long and could be split for clarity.
- Page 4: "Brazil has no national epidemiological surveillance system for B19V, and cases are not compulsorily reported." This sentence is repeated multiple times.
- Page 7: "The overall detection of B19V DNA in our study was lower (6.1%), than showed in previous studies" should be "The overall detection of B19V DNA in our study was lower (6.1%) than that reported in previous studies."
- Page 10: "B19V DNA detection was significantly higher (OR = 28.1, CI = 13.5–58.5, p = 0.001) in the CKD patients under dialysis (65%) than in the control group (blood donors; 6.3%), because of the immune deficiency condition in these patients and the routine exposure to dialysis procedures, and therefore there is a potential risk for parenterally transmitted infections." This sentence is overly long and could be split for clarity.
Comments on the Quality of English LanguageThe English could be improved to more clearly express the research.
Author Response
Comment: This retrospective analysis investigates the clinical and epidemiological aspects of Parvovirus B19 (B19V) infection in Brazil, focusing on both immunocompetent and immunocompromised individuals. B19V, a widespread human virus transmitted primarily through respiratory routes, is often associated with mild, self-limiting illnesses such as rash diseases and acute febrile illness. However, it can also lead to severe complications, particularly in high-risk groups like pregnant women, individuals with hematological disorders, and those with compromised immune systems. Despite its potential for serious health impacts, B19V remains underdiagnosed and underreported, often overshadowed by other epidemic viruses such as dengue, Zika, and chikungunya.
The study highlights the importance of including B19V in differential diagnoses, especially in cases of rash diseases, acute febrile illness, chronic kidney disease (CKD), HIV co-infection, and acute liver failure (ALF). It underscores the need for enhanced surveillance and diagnostic strategies to mitigate the clinical impact of B19V, particularly in Brazil, where the virus exhibits cyclical outbreaks and regional variations in prevalence. The research also explores the molecular epidemiology of B19V, identifying the predominance of genotype 1 in Brazil, with occasional detection of genotype 3b, which has been associated with more severe outcomes.
Specific comments:
Abstract:
The abstract provides a good summary of the study's objectives, methods, and key findings. However, it is somewhat lengthy and could be more concise. For instance, the phrase "by considering to include B19V in differential laboratory diagnosis" is awkwardly phrased and should be reworded for clarity. Additionally, the abstract does not mention the limitations of the study, which is important for setting expectations. Including a brief statement about the study's limitations, such as the lack of national surveillance data or the retrospective nature of the analysis, would provide a more balanced overview.
Reply: Thank you for pointing this out. The mentioned phrase was reworded and stated in line 21 as: “…to include B19V in the routine of diagnosis…”. The limitation of the study was stated in the final phrase of the abstract (lines 22-23) as: “This study was limited by the absence of national surveillance data of B19V in Brazil and by the analyses that occurred retrospectively.”
Comment: Introduction:
The introduction effectively sets the stage for the study by discussing the transmission, clinical manifestations, and risks associated with B19V infection. However, it could be improved by more clearly identifying the gaps in the literature, particularly in the context of Brazil. For example, the introduction could emphasize the lack of recent data on B19V prevalence in Brazil and the need for updated epidemiological studies. Additionally, the phrase "Despite the infection is general self-limited" contains a grammatical error ("is general" should be "is generally"). The introduction also lacks a clear hypothesis or research question, which would help guide the reader and provide a stronger foundation for the study.
Reply: In lines 97-99, the following phrase was added: “Therefore, it is possible to note the absence of recent data on the prevalence of B19V in the Brazilian population, which is urgently demanding updated epidemiological studies.” to reinforce the lack of recent data on B19V prevalence in Brazil. The phrase stated in line 39 was typographically corrected. And the hypothesis was added in lines 102-104: “The hypothesis of this review is the need to include B19V in the differential diagnosis of viral infections in specific population groups, as it is an important etiological agent of severe anemia.”
Comment: Epidemiology of Parvovirus B19 in Brazil:
- The section is somewhat repetitive, particularly in discussing the cyclical nature of outbreaks.
Reply: Thank you for pointing this out. In this newly submitted version, we removed the phrases in places we thought were repetitive.
Comment: - The phrase "Brazil has no national epidemiological surveillance system for B19V, and cases are not compulsorily reported" is repeated multiple times, which could be streamlined.
Reply: Thank you for pointing this out. In this newly submitted version, we removed the phrases in places we thought were repetitive.
Comment: - The discussion on the impact of COVID-19 on B19V epidemiology is interesting but lacks depth. More data or references could be provided to support the claims.
Reply: Thank you for pointing this out. In this newly submitted version, a new paragraph with new references regarding on this topic was added on lines 117-122:
Lines 117-122: The re-emergence of B19V after the COVID-19 pandemic, at a level equal to or higher than the pre-pandemic period, has been described in several countries such as France [29], the Netherlands [30], Serbia [31], and the United States [32]. This is due to a decline in population immunity due to the social distancing required during the pandemic, creating a pool of susceptible individuals [33].
29. d'Humières C, Fouillet A, Verdurme L, Lakoussan SB, Gallien Y, Coignard C, Hervo M, Ebel A, Soares A, Visseaux B, Maire B, Juan PH, Parent du Châtelet I, Guthmann JP, Durand J. An unusual outbreak of parvovirus B19 infections, France, 2023 to 2024. Euro Surveill. 2024, 29(25):2400339.
30. Russcher A, van Boven M, Benincà E, Verweij EJTJ, Molenaar-de Backer MWA, Zaaijer HL, Vossen ACTM, Kroes ACM. Changing epidemiology of parvovirus B19 in the Netherlands since 1990, including its re-emergence after the COVID-19 pandemic. Sci Rep. 2024, 26;14(1):9630.
31. Vuković V, Patić A, Ristić M, Kovačević G, Hrnjaković Cvjetković I, Petrović V. Seroepidemiology of Human Parvovirus B19 Infection among the Population of Vojvodina, Serbia, over a 16-Year Period (2008-2023). Viruses. 2024, 25;16(2):180.
32. Alfego D, Hernandez-Romieu AC, Briggs-Hagen M, Dietz S, Gillim L, Dale SE, Grover A, Albrecht J, Sesok-Pizzini D, Ei-senberg M, Gregory CO, Poirier B. Detection of Increased Activity of Human Parvovirus B19 Using Commercial Laboratory Testing of Clinical Samples and Source Plasma Donor Pools - United States, 2024. MMWR Morb Mortal Wkly Rep. 2024, 28;73(47):1076-1081.
33. Giovanetti M, Branda F, Scarpa F, Ciccozzi M, Ceccarelli G. Letter to the editor: Severe parvovirus B19 infections in the immunocompetent population. Euro Surveill. 2024, 29(29):2400438.
Comment: - The section could benefit from a clearer structure, perhaps with subheadings to separate discussions on prevalence, seasonal patterns, and regional differences.
Reply: Thank you for pointing this out. In this newly submitted version, we added three subheadins: 2.1. Seasonal pattern of the infection, 2.2. Prevalence of B19V infection, and 2.3. Molecular epidemiology of B19V infection.
Comment: Parvovirus B19 Association to Typical and Uncommon Clinical Manifestations:
- The section is overly long and could be divided into subsections for better readability.
Reply: Thank you for pointing this out. In this newly submitted version, we divided the section 3. Parvovirus B19 association with typical and uncommon clinical manifestations in 3. Parvovirus B19 association with typical clinical manifestations (line 212) and 4. 4. Parvovirus B19 association with uncommon clinical manifestations (line 275).
Comment: - The discussion on rash diseases and acute febrile illness is somewhat repetitive, particularly in the context of anemia.
Reply: Thank you for pointing this out. In this newly submitted version, we reduced the discussion about anemia in RD and AFI patients.
Comment: - The phrase "The overall detection of B19V DNA in our study was lower (6.1%), than showed in previous studies" is awkwardly phrased and should be reworded.
Reply: Thank you for pointing this out. In this newly submitted version, we reworded the phrase as stated in line 249 “than that reported in previous studies:”.
Comment: - The discussion on chronic kidney disease lacks a clear conclusion on the clinical significance of persistent B19V infection in these patients.
Reply: Thank you for pointing this out. In this newly submitted version, we rewrote the conclusion of this topic to improve the understanding of the text.
Comment: - The section on HIV co-infection is informative but could benefit from more recent data, as the cited studies are from 2001-2003.
Reply: Thank you for pointing this out. Unfortunately, we cannot add more recent information, as there is no published data on this population in Brazil after the period mentioned.
Comment: - The discussion on acute liver failure is thorough but could be more concise. The mechanism of liver damage by B19V is discussed in detail, but the clinical relevance of these findings could be emphasized more.
Reply: Thank you for pointing this out. In this newly submitted version, this paragraph was emphasized and stated as follow (lines 400-401):
Lines 400-401: B19V contributed to liver damage among patients with acute liver failure; clinically, the worse outcomes were for patients co-infected with HBV.
Comment: Conclusion:
The conclusion effectively summarizes the key findings and emphasizes the importance of B19V in differential diagnosis. However, it is somewhat repetitive, particularly in reiterating the need for enhanced surveillance. The phrase "underscore the importance of including B19V in the differential diagnosis" is awkwardly phrased and should be reworded. Additionally, the conclusion could benefit from a more forward-looking perspective. For example, it could discuss potential future research directions, such as longitudinal studies to establish causal relationships between B19V infection and clinical outcomes, or public health interventions to improve surveillance and diagnosis.
Reply: Thank you for pointing this out. In this newly submitted version, the conclusion was completely rewritted, as stated in lines: 406-432:
Lines 406-432: These results highlight the potential role of B19V as a causative agent associated with various clinical manifestations and the need for future longitudinal studies to establish causal relationships between B19V infection and clinical outcomes. The prevalence identified in this study suggests that B19V is present in different population groups and emphasizes the importance of enhanced surveillance of this infection, as it affects both im-immunocompetent and immunocompromised individuals. Therefore, including B19V in the differential diagnosis is crucial for epidemiological purposes and effective patient management.
Given these considerations, Brazil needs to enhance its surveillance efforts to monitor the epidemiology of B19V. This will help in promptly identifying potential outbreaks and tracking case reports, particularly among individuals at higher risk of complications. We recommend employing strategies similar to those that have been implemented in several European countries. For instance, all blood donations in Germany and four regions in Austria are screened using a real-time nucleic acid test (NAT) for B19V [146,147]. Additionally, since 2008, Japanese Red Blood Centers have been using a chemiluminescent enzyme immunoassay to screen all donated blood for B19V antigens [148]. Furthermore, it is essential to notify the cases of B19V infection in the Brazil´s database of the Notifiable Diseases Information System – SINAN. This system relies on medical reports of diseases that require mandatory notification. The SINAN database is a valuable of information for national assessment of epidemiological Surveillance assessment. However, the B19V infraction is not currently included in the list of compulsory notifications. Fully utilizing this system could enable the collection of crucial data for calculating key indicators needed to monitor this infection, and support the formulation and evaluation of health policies, plans, and programs, supporting the decision-making process and ultimately contributing to improving the health of the population. Also, is important the occurrence of longitudinal studies to establish causal relationships between B19V infection and clinical outcomes, in all previously studied populations reported.
Comment: Figures:
Figure 3
- The tree is cluttered and difficult to read, particularly due to the large number of sequences and the small font size. The labels for the sequences are not clearly differentiated, making it hard to distinguish between different groups (e.g., acute liver failure, chronic kidney disease, etc.).
- The bootstrap values are not clearly indicated, which is crucial for assessing the robustness of the tree's branches. Without these values, the reliability of the phylogenetic relationships is questionable.
- The tree does not clearly differentiate between the different genotypes (1a, 1b, 2, 3a, 3b). A color-coded or labeled system could help in distinguishing these genotypes more effectively.
- The use of human bocavirus (HBoV) as an outgroup is appropriate, but the tree does not clearly show the divergence between B19V and HBoV, which could be improved by adjusting the scale or adding more outgroup sequences.
- The tree lacks clear annotations for key nodes or clades, which would help in interpreting the evolutionary relationships between the sequences.
- The manuscript does not provide sufficient details on the methodology used to construct the tree (e.g., alignment method, model selection, etc.). This information is crucial for reproducibility and validation of the results.
Reply: The phylogenetic tree was completely redone, with fewer prototype sequences and a larger font for better reading of the sequences used. The sequence labels were colored to improve group differentiation. The bootstrap values are on the left side of each branch of the tree, as MEGA software usually maintains. The genotypes (1a, 1b, 2, 3a and 3b) were colored on the tree branches and the color information was added to the figure legend (Lines 268-275). We adjusted the scale of the tree root using the HBoV sequence. Since the manuscript is a review, there is no topic for the methodology, but the figure legend contains all the necessary information for the phylogenetic tree. We also added 2 references to improve the understanding and reproducibility of the result contained in the phylogenetic tree:
115. Tamura K. and Nei M. Estimation of the number of nucleotide substitutions in the control region of mitochondrial DNA in humans and chimpanzees. Molecular Biology and Evolution. 1993, 10:512-526.
116. Felsenstein J. Confidence limits on phylogenies: An approach using the bootstrap. Evolution. 1985, 39:783-791.
Comment: Typographical Errors:
- Page 1: "Despite the infection is general self-limited" should be "Despite the infection being generally self-limited."
- Page 2: "transient aplastic crisis (TAC). TAC was the first human illness associated with B19V" is repetitive. Consider rephrasing.
- Page 3: "correlating molecular and serological tests for an accurate laboratory diagnosis of B19V infection [21] due to the possibility of false-negative results since antibodies against the viral proteins may be complexed with viral particles and consequently become undetectable in serological assays [22]." This sentence is overly long and could be split for clarity.
- Page 4: "Brazil has no national epidemiological surveillance system for B19V, and cases are not compulsorily reported." This sentence is repeated multiple times.
- Page 7: "The overall detection of B19V DNA in our study was lower (6.1%), than showed in previous studies" should be "The overall detection of B19V DNA in our study was lower (6.1%) than that reported in previous studies."
- Page 10: "B19V DNA detection was significantly higher (OR = 28.1, CI = 13.5–58.5, p = 0.001) in the CKD patients under dialysis (65%) than in the control group (blood donors; 6.3%), because of the immune deficiency condition in these patients and the routine exposure to dialysis procedures, and therefore there is a potential risk for parenterally transmitted infections." This sentence is overly long and could be split for clarity.
Reply: Thank you for pointing this out. All typographical errors have been corrected in the newly submitted version of the review.

Round 2
Reviewer 1 Report
Comments and Suggestions for Authors
Fix syntax in Conclusion:
-The SINAN database is a valuable source of information for national epidemiological surveillance assessment.
- Also important is the occurrence....
Comments on the Quality of English LanguageSee comments to editor. Fix syntax in the Conclusion section.
Author Response
Comment: Fix syntax in Conclusion:
-The SINAN database is a valuable source of information for national epidemiological surveillance assessment.
- Also important is the occurrence....
Reply: Thank you for pointing this out. All sentences were corrected in this version of the manuscript.
